# Improved Sample Complexity for Full Coverage in Compact and Continuous Spaces via Probabilistic Analysis

## Abstract

Verifying uniform conditions over continuous spaces through random sampling is fundamental in machine learning and control theory, yet classical coverage analyses often yield conservative bounds, particularly at small failure probabilities. We study uniform random sampling on the $d$-dimensional unit hypercube and analyze the number of uncovered subcubes after discretization. By applying a concentration inequality to the uncovered-count statistic, we derive a sample complexity bound with a logarithmic dependence on the failure probability ($\delta$), i.e., $M = O(\tilde{C}\ln(\frac{2\tilde{C}}{\delta}))$, which contrasts sharply with the classical linear $1/\delta$ dependence. Under standard Lipschitz and uniformity assumptions, we present a self-contained derivation and compare our result with classical coupon-collector rates. Numerical studies across dimensions, precision levels, and confidence targets indicate that our bound tracks practical coverage requirements more tightly and scales favorably as $\delta \to 0$. Our findings offer a sharper theoretical tool for algorithms that rely on grid-based coverage guarantees, enabling more efficient sampling, especially in high-confidence regimes.

## 1 Introduction

Many algorithms in machine learning and control theory rely on theoretical conditions that must hold uniformly across continuous, compact spaces (Hornik et al., 1989; Lillicrap et al., 2015; Antos et al., 2007; Bensoussan et al., 2022; Jia & Zhou, 2022; Tabuada & Gharesifard, 2020). Verifying such conditions at every point is infeasible, as it would require checking an uncountable set. This challenge has motivated statistical frameworks that discretize compact domains into grid cells and cast the verification task as a variant of the coupon collector problem (Motwani & Raghavan, 1996; Younes & Simmons, 2002; Kakade, 2003).

A concrete and important example of this challenge arose in the context of Deep Contraction Drift Calculators (DCDC) for solving Contraction Drift Equations (CDEs) (Qu et al., 2024). This algorithm requires a Contraction Drift (CD) condition to be satisfied throughout the continuous state space $\mathcal{X}$ for solution validity. In the theoretical analysis of such algorithms, a critical step involves determining sample sizes $M$ and $N$ to ensure that a probability inequality of the form

$$P\left(\sup_{x \in \mathcal{X}}\big[KV(x) - V(x) + U(x)\big] \le \sup_{x \in \mathcal{M}}\big[\hat{K}_N V(x) - V(x) + U(x)\big] + \varepsilon/2\right) > 1 - \delta,$$

where $\varepsilon > 0$ is the target tolerance and $\delta \in (0, 1)$ is the failure probability, holds with high confidence. Here, $\mathcal{X}$ represents the continuous state space, $\mathcal{M}$ denotes a finite sample set, and the functions $KV$, $V$ and $U$ encode the contraction drift condition essential for algorithm convergence.

The core challenge in such problems lies in bridging the gap between the supremum over the entire continuous space $\mathcal{X}$ and the supremum over a finite sample set $\mathcal{M}$ (Shalev-Shwartz & Ben-David, 2014). Specifically, we aim to bound the probability that

$$P\left(\sup_{x \in \mathcal{X}}\big[KV(x) - V(x) + U(x)\big] > \sup_{x \in \mathcal{M}}\big[KV(x) - V(x) + U(x)\big] + \varepsilon/2\right) \le \frac{\delta}{2}.$$

This probability constraint ensures that the finite sample approximation is sufficiently accurate for practical algorithm implementation (Webb et al., 2018; Caflisch, 1998; Mey & Loog, 2021).

Verifying these conditions pointwise is infeasible, as it requires checking an uncountable set. A common strategy, therefore, is to sample $M$ discrete points uniformly at random (Cochran, 1977; Metropolis & Ulam, 1949; Caflisch, 1998; Calafiore & Campi, 2006; Valiant, 1984; Vapnik, 2013) from the space $\mathcal{X}$ and verify the condition on this finite set (Qu et al., 2024). If the desired condition holds for all $M$ sample points, it is inferred with some controlled probability that the condition holds over the entire space (Fournier et al., 2011; Valiant, 1984; Vapnik, 2013; Tem, 2005). This immediately raises the critical research question: what is the minimum number of samples $M$ (the sample complexity) needed to "cover" the space adequately, ensuring the reliability of this approximation?

The sample complexity $M$ is often linked to the classical coupon collector problem (Motwani & Raghavan, 1996; Ferrante & Saltalamacchia, 2014; Azimi et al., 2017). Here, the continuous domain is discretized into $\tilde{C}$ subregions ("coupons"), with $\tilde{C}$ typically scaling as $O(\varepsilon^{-d})$ for a precision parameter $\varepsilon$. Standard analyses show an expected requirement of $\Theta(\tilde{C} \ln \tilde{C})$ (Motwani & Raghavan, 1996; Ferrante & Saltalamacchia, 2014). When considering a failure probability $\delta$ (risk of incomplete coverage), analyses using Markov's inequality typically yield sample complexity bounds of $M = O(\frac{\tilde{C} \ln \tilde{C}}{\delta})$.

However, a significant deficiency of these existing bounds is their linear $1/\delta$ dependence on the failure probability. This means that requiring higher confidence dramatically increases the estimated sample size $M$, often leading to overly conservative estimates. For instance, in the DCDC algorithm context(Qu et al., 2024), when dealing with functions satisfying Lipschitz conditions with constant $\tilde{L}$, the classical approach often overestimates the required sample size, potentially hindering practical implementation in resource-constrained scenarios.

The primary objective of this research is to address this gap by deriving a more precise sample complexity bound for achieving full coverage in compact spaces via uniform random sampling, specifically aiming for an improved dependency on the failure probability $\delta$. Our study employs a direct probabilistic analysis of uncovered sub-regions after space discretization, leveraging Chebyshev's inequality (Feller, 1991) rather than the classical Markov bound. The scope is focused on uniform random sampling within a $d$-dimensional unit hypercube $[0,1]^d$, with applications to problems like the DCDC verification task.

Our analysis establishes a sample complexity of $M = O(\tilde{C} \ln(\frac{2\tilde{C}}{\delta}))$, offering a more favorable $\ln(1/\delta)$ dependency compared to the classical $1/\delta$ scaling. This improvement is particularly significant for scenarios requiring high confidence (small $\delta$) or addressing high-precision problems. The result is further substantiated by comprehensive numerical experiments across various parameter regimes.

Our contributions are threefold:

- We derive a sample complexity bound with logarithmic dependence on the failure probability, $M = O(\tilde{C} \ln(2\tilde{C}/\delta))$, by analyzing the uncovered-subcube count under uniform sampling and standard Lipschitz assumptions.

- We contrast this result with classical coupon-collector-based rates exhibiting linear $1/\delta$ dependence, clarifying when and why the proposed bound yields tighter coverage requirements.

- We validate our theoretical claims through a tripartite numerical study. Across a direct grid-coverage simulation, a deep learning model for DCDC algorithm, and a real-world housing price regression, we analyze the impact of varying data dimension, precision, and confidence. The findings consistently indicate that our bound more closely reflects the actual sample sizes required for task success.

The remainder of this paper is organized as follows. Section 2 introduces the necessary mathematical preliminaries, outlines core assumptions, and presents our main theoretical results. Section 3 describes the numerical studies conducted to validate our theoretical findings, including the experimental setup and analysis of parameter impacts, and discusses the implications of our results and compares them with classical approaches. Finally, Section 4 concludes the paper by summarizing the main contributions, discussing

limitations, and suggesting directions for future work. The appendix provides a detailed proof of Theorem 1.

## 2 Related Work

Coverage analysis in discrete settings traces back to classical occupancy and coupon collector problems (Motwani & Raghavan, 1996; Ferrante & Saltalamacchia, 2014). In learning theory and Monte Carlo methods, uniform guarantees over discretized spaces are typically established through union bounds or metric entropy arguments (Shalev-Shwartz & Ben-David, 2014; Caflisch, 1998; Webb et al., 2018). In high dimensions, Lipschitz structure and metric entropy guide discretization complexity (Valiant, 1984; Fournier et al., 2011). Our work focuses on the uncovered-count statistic and obtains a bound with improved $\ln(1/\delta)$ dependence under uniform sampling; it complements classical expectation-based arguments and aligns with concentration-based perspectives (Feller, 1991). We also connect to verification-style use cases in control-inspired learning where uniform conditions drive correctness (Mey & Loog, 2021; Qu et al., 2024).

## 3 Preliminaries and Main Results

### 3.1 Mathematical Foundations and Core Assumptions

Based on the motivating example discussed in the Introduction, we now formalize the mathematical framework for our analysis. Consider the problem of random sampling on a compact space $\mathcal{X} = [0,1]^d$. Our goal is to determine the sample size $M$ such that the probability inequality

$$P\left(\sup_{x \in \mathcal{X}} \left[KV(x) - V(x) + U(x)\right] > \sup_{x \in \mathcal{M}} \left[KV(x) - V(x) + U(x)\right] + \varepsilon/2\right) \leq \frac{\delta}{2}$$

holds with high confidence, where $\mathcal{M}$ represents a finite random sample set of size $M$. To address this problem systematically, we must analyze the Lipschitz properties of the function $KV - V + U$. For any $x, y \in \mathcal{X}$, the Lipschitz analysis yields:

$$\begin{aligned} |KV(x) - KV(y)| &= |\mathbb{E}Df(x)V(f(x)) - \mathbb{E}Df(y)V(f(y))| \\ &\leq |\mathbb{E}Df(x)V(f(x)) - \mathbb{E}Df(x)V(f(y))| + |\mathbb{E}Df(x)V(f(y)) - \mathbb{E}Df(y)V(f(y))| \\ &\leq \mathbb{E}Df(x)|V(f(x)) - V(f(y))| + \mathbb{E}|Df(x) - Df(y)|V(f(y)) \\ &\leq DV \cdot \mathbb{E}Df^2 \cdot \|x - y\| + \sup V \cdot \mathbb{E}D^2f \cdot \|x - y\|, \end{aligned}$$

where $DV$ denotes the Lipschitz constant of the function $V$. Therefore, the Lipschitz constant $\tilde{L}$ of the function $KV - V + U$ satisfies:

$$\tilde{L} \overset{\text{def}}{=} DV \cdot \mathbb{E}Df^2 + \sup V \cdot \mathbb{E}D^2f + DV + DU.$$

Without loss of generality, let the research space be a $d$-dimensional unit cube $X = [0,1]^d$. We discretize the space by uniformly dividing each dimension into $k$ segments. To ensure the function variation within any sub-region is bounded, we set $k = \lceil 2\tilde{L}\sqrt{d}/\varepsilon \rceil$. The parameter $\tilde{r}$ is defined as $\tilde{r} = \frac{\varepsilon}{2\tilde{L}}$, where $\varepsilon$ is the precision parameter and $\tilde{L}$ is the Lipschitz constant. The total number of subcubes $\tilde{C}$, is thus given by:

$$\tilde{C} = k^d = \left(\left\lceil \frac{2\tilde{L}\sqrt{d}}{\varepsilon} \right\rceil\right)^d = O(\varepsilon^{-d}). \tag{1}$$

Let $Z$ be the number of subcubes that are not covered after $M$ independent random samples. The coverage failure event is $Z \geq 1$. For each subcube $i$, define the indicator random variable $I_{i,j}$:

$$I_{i,j} = \begin{cases} 1, & \text{the j-th sample does not cover subcube i ;} \\ 0, & \text{the j-th sample covers subcube i.} \end{cases}$$

Assuming that the sample points are uniformly distributed in $[0, 1]^d$, the probability that a particular subcube $i$ is covered by a single sample is $p_s = 1/\tilde{C}$. Let $Y_i$ denote the indicator variable that subcube $i$ is not covered after $M$ samples:

$$Y_i = \prod_{j=1}^{M} I_{i,j}.$$

The total number of uncovered subcubes $Z$ can be expressed as:

$$Z = \sum_{i=1}^{\tilde{C}} Y_i. \tag{2}$$

The following core assumptions underpin this study:

1. The sample points are independently and identically distributed (i.i.d.), following a uniform distribution on $[0, 1]^d$.

2. The space $[0, 1]^d$ can be divided into $\tilde{C}$ equal volume subcubes.

3. The probability of a single sampling covering any subcube is equal, that is, $p_s = 1/\tilde{C}$.

### 3.2 Main Theorems and Corollaries

**Lemma 1**:(Expectation of the number of uncovered subcubes)

The expected value of the number of uncovered subcubes $Z$ after $M$ independent samples is:

$$E[Z] = \tilde{C}\left(1 - \frac{1}{\tilde{C}}\right)^M. \tag{3}$$

*Proof.* : For any subcube $i$, the probability of being uncovered after $M$ samples is:

$$E[Y_i] = P(Y_i = 1) = \left(1 - \frac{1}{\tilde{C}}\right)^M. \tag{4}$$

$\square$

By the linearity of expectation, combining equations (2) and (4), we obtain:

$$E[Z] = \sum_{i=1}^{\tilde{C}} E[Y_i] = \tilde{C}\left(1 - \frac{1}{\tilde{C}}\right)^M.$$

**Theorem 1** (Sample Complexity with Logarithmic Dependence). *Suppose $\mathcal{X} = [0, 1]^d$ is discretized into $\tilde{C}$ equal subcubes, and the sampling distribution is uniform over $\mathcal{X}$. Under the assumed Lipschitz continuity of $V$ and $U$ (with discretization resolution chosen accordingly), to achieve full coverage with probability at least $1 - \delta$, it suffices that*

$$M = O(\tilde{C}\ln\frac{2\tilde{C}}{\delta}).$$

*A more precise lower bound is given by the following formula:*

$$M = O(\frac{\ln q_1}{\ln(1 - \frac{1}{\tilde{C}})}),$$

*where $q_1 = \frac{2\tilde{C}(1+\delta) - \sqrt{4\tilde{C}^2 + 8\tilde{C}\delta(\tilde{C}-1)}}{2(2\tilde{C} + \delta\tilde{C}^2)}$.*

Comparing with the classical method, the classic coupon collection model gives $M_2 = O(\tilde{C}\ln\tilde{C}/\delta)$.

The method in this paper yields $M_1 = O(\tilde{C}\ln\frac{2\tilde{C}}{\delta})$. Our bound replaces the linear dependence on $1/\delta$ with a much milder logarithmic dependence$\ln(1/\delta)$, clarifying a potentially substantial improvement in high-confidence regimes.

## 4    Numerical Studies

This section presents a series of numerical experiments designed to validate the theoretical sample complexity bounds derived in Section 2. We conduct a comprehensive parameter analysis to assess the performance of our proposed model against classical theories under various conditions.

### 4.1    Validation of Theoretical Sample Complexity

This section presents a series of numerical experiments designed to validate the theoretical sample complexity bounds derived in Section 2. We conduct a comprehensive parameter analysis to assess the performance of our proposed model against classical theories under various conditions. The benchmark parameters for this study are set as follows: dimension $d = 2$, precision $\varepsilon = 0.1$, failure probability $\delta = 0.1$, and Lipschitz constant $\tilde{L} = 1.0$. When analyzing one parameter, the others are held at their benchmark values. For each parameter configuration, we performed 32 independent trials to ensure statistical robustness. In the first experiment, we analyze the effect of dimension $d \in \{1, 2, 3\}$. The results are presented in Figure 1. The left plot shows that the ratio of actual samples to our theoretical bound ($M_{actual}/M_{theory}$) remains consistently around an average of 0.72, indicating a tight and accurate prediction. The center plot starkly contrasts our theoretical sample size requirement with the classic method's, showing that the classic bound grows much more rapidly with dimension. The rightmost plot quantifies this advantage, indicating an improvement of over 80% across all tested dimensions, reaching up to 87.6% for $d = 3$.

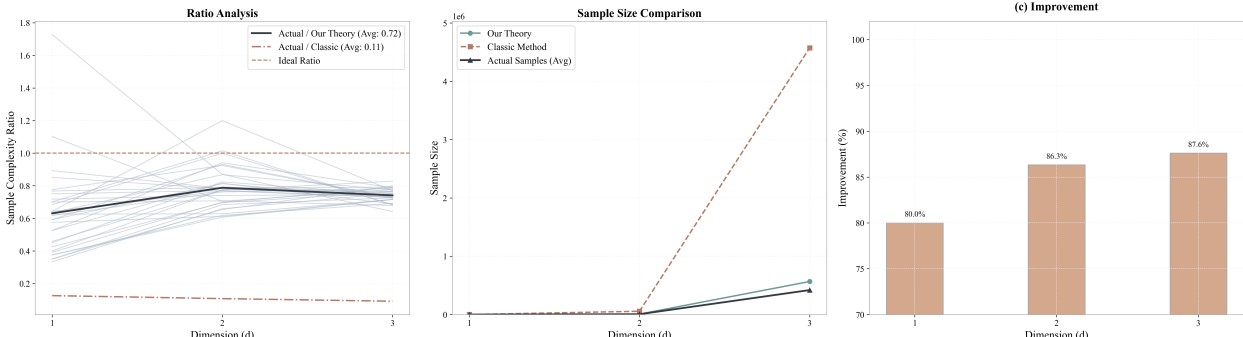

Figure 1: Analysis of Dimension Parameter (d). Subplots show: (a) Ratio of actual to theoretical sample complexity, (b) Comparison of theoretical and actual sample sizes, (c) Improvement of our method over the classic method.

The second experiment details the analysis for precision $\varepsilon \in [0.05, 0.5]$, with results shown in Figure 2. The ratio analysis (left plot) shows that our theoretical bound remains a strong predictor across the entire range of $\varepsilon$ values, with the average ratio fluctuating around 0.80. The theoretical comparison (center plot) reveals that the sample size required by the classic method increases dramatically for small $\varepsilon$, whereas our method requires a much more manageable number of samples. The improvement plot (right) confirms a consistent performance gain of over 85% for our method.

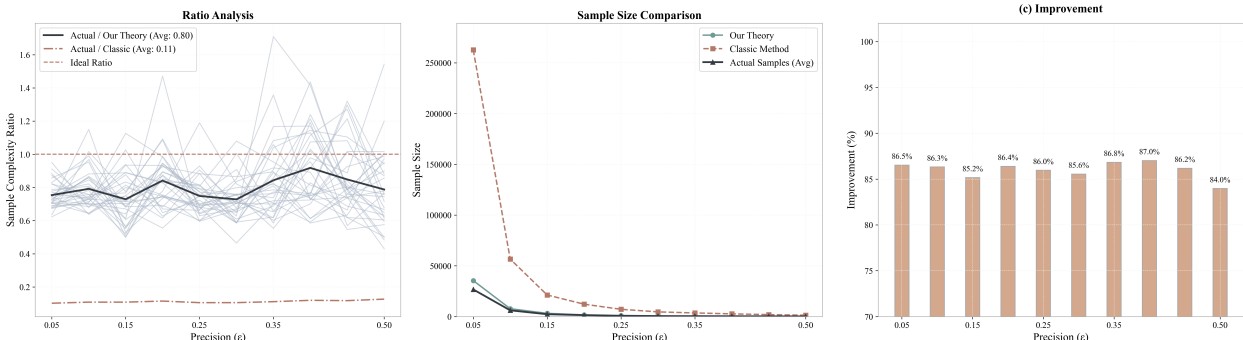

Figure 2: Analysis of Precision Parameter ($\varepsilon$). Subplots show: (a) Ratio of actual to theoretical sample complexity, (b) Comparison of theoretical and actual sample sizes, (c) Improvement of our method over the classic method.

The third experiment examines the impact of the failure probability $\delta \in [0.02, 0.2]$, as illustrated in Figure 3. The ratio plot (left) confirms that our model's predictions are robust, with the $M_{actual}/M_{theory}$ ratio staying close to an average of 0.77. The center plot highlights the practical benefit of our model's logarithmic dependence on $1/\delta$. As $\delta$ approaches zero, the sample size required by the classic method becomes prohibitively large, while our proposed requirement grows much more slowly. The improvement plot (right) shows that the advantage of our method is most pronounced at high-confidence (low $\delta$) requirements, with improvements ranging from 74.6% to over 95.9%.

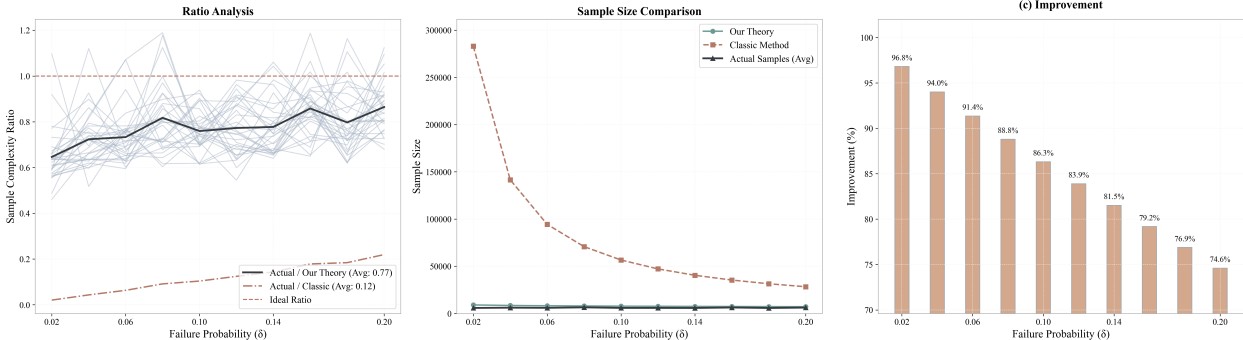

Figure 3: Analysis of Failure Probability Parameter ($\delta$). Subplots show: (a) Ratio of actual to theoretical sample complexity, (b) Comparison of theoretical and actual sample sizes, (c) Improvement of our method over the classic method.

In summary, the numerical results across all three parameter studies robustly validate our theoretical findings. Our proposed sample complexity bound is not only significantly tighter and more practical than classical bounds but also accurately predicts the actual sample requirements observed in simulations.

## 4.2 Application in DCDC Framework

To further validate the practical utility of our theoretical bounds, we conducted an experiment within the Deep Contraction Drift Calculators (DCDC) framework, applying it to a 2D Ornstein-Uhlenbeck process. The experiment was configured with a precision of $\varepsilon = 0.05$ and a failure probability of $\delta = 0.01$.

Based on these parameters, our theoretical sample size was calculated to be $M_{theory} = 43,486$, whereas the classical method yielded a much larger requirement of $M_{classic} = 2,852,379$. We established three experimental groups: an "Insufficient" group with half of our theoretical sample size, a "Sufficient" group using our full theoretical sample size, and a "Conservative" group using the classical sample size.

The primary results, summarized in Figure 4, demonstrate that the "Sufficient" group achieved a CDE satisfaction rate of 99.19%, which is comparable to the 99.15% rate of the "Conservative" group, despite using approximately 98.5% fewer samples. This outcome strongly supports our theory's effectiveness in a real-world application, confirming that it provides a reliable and significantly more efficient sample complexity bound.

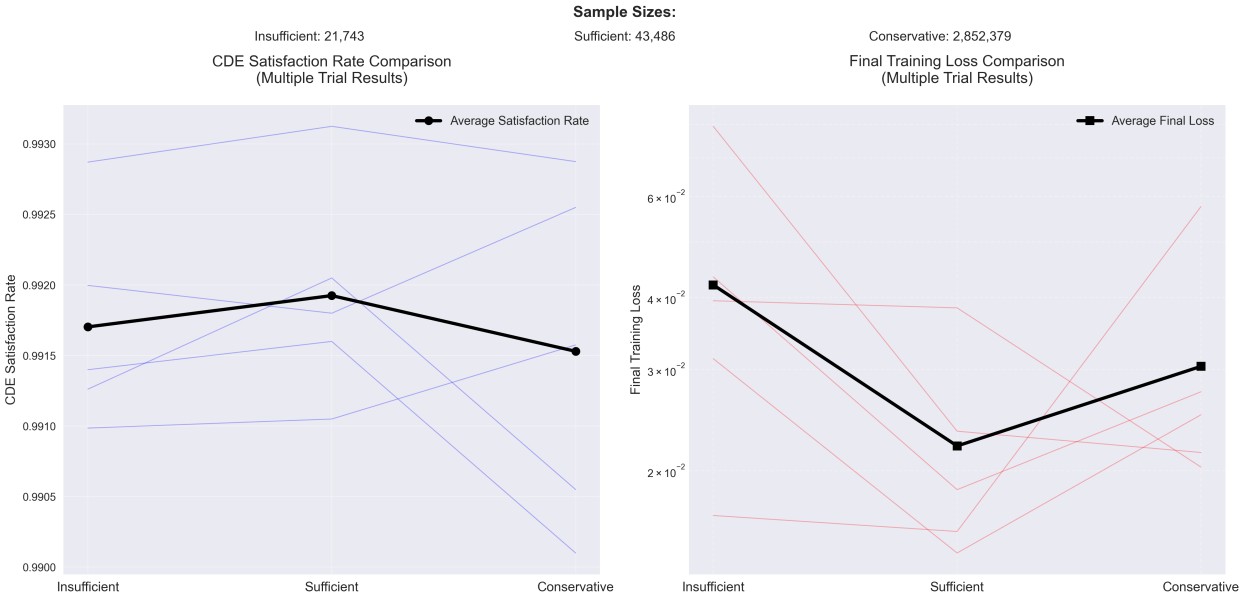

Figure 4: Validation of sample complexity in the DCDC framework. The chart displays the CDE satisfaction rate and final training loss for the Insufficient, Sufficient, and Conservative sample groups.

The training dynamics are detailed in Figure 5, which plots the average training loss on a logarithmic scale. All three groups show robust convergence, characterized by a rapid initial decrease in loss followed by stabilization. The "Sufficient" and "Conservative" groups achieve nearly indistinguishable final loss values, demonstrating that the sample size derived from our theory is adequate for effective training. While the "Insufficient" group also converges, its final loss is slightly elevated, which is consistent with its lower CDE satisfaction rate. The tight and overlapping variance bands across all groups highlight the stability of the learning process.

Furthermore, Figure 6 visualizes the learned Lyapunov-like function (V-function) and the CDE residual surfaces, averaged over five trials. All three groups learned the expected bowl-shaped V-function, indicative of a stable system. The residual plots confirm that the CDE condition ($KV - V < 0$) was satisfied across most of the state space, with minor positive residuals near the origin, which is an expected artifact of the learning process. The similarity of the surfaces between the "Sufficient" and "Conservative" groups further reinforces that our sample complexity bound is adequate for learning the correct system dynamics.

## 4.3 Empirical Validation of Sample-Complexity Bounds on the California Housing Dataset

To empirically validate the theoretical sample complexity bounds derived in this paper, we conducted a series of experiments on the California Housing dataset. This dataset is a classic regression problem, well-suited for testing our framework's ability to predict performance and guide sample size selection in a real-world scenario.

The primary objective of the experiment is to verify whether the model's performance on the test set aligns with the predictions of our theoretical bounds. We designed the experiment around three distinct groups of training sample sizes, determined by our theoretical calculations:

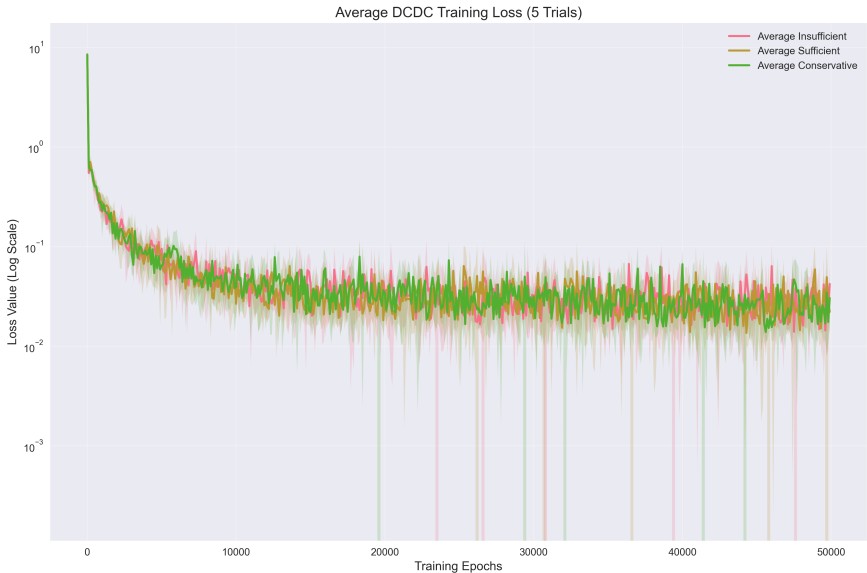

Figure 5: Average DCDC training loss over 5 trials. The plot shows the convergence of the training process for each sample group on a log scale.

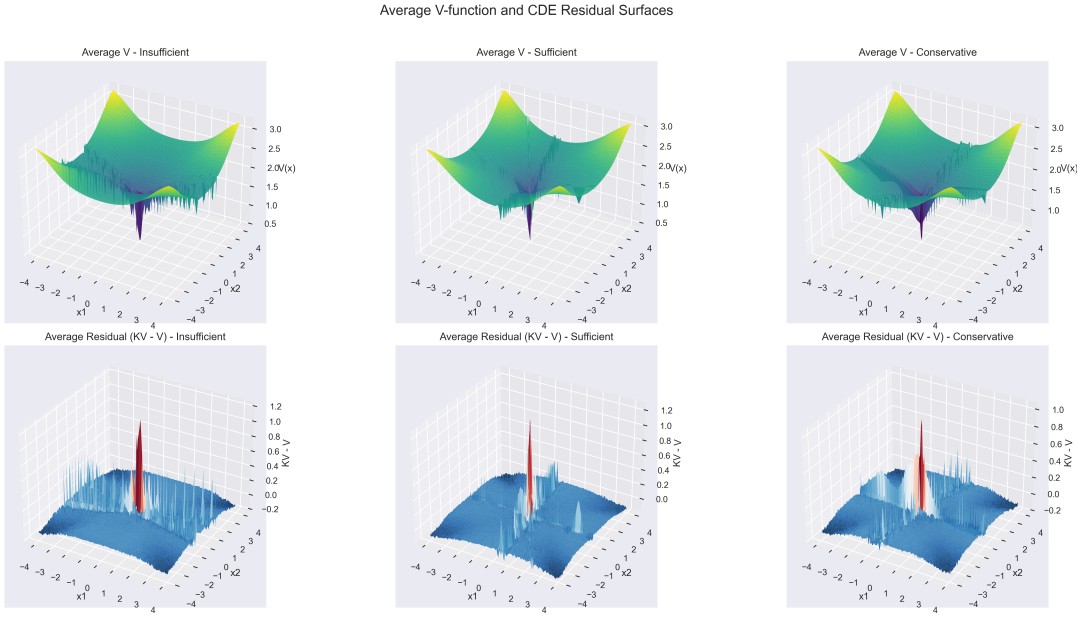

Figure 6: Average V-function and CDE residual surfaces. The top row shows the learned V-function, and the bottom row shows the CDE residual $(KV - V)$ for each sample group.

- **Insufficient Group:** A sample size significantly smaller than our calculated sufficient bound ($0.1 \times M_1$), intended to represent a scenario where the data is inadequate for achieving the desired performance.

- **Sufficient Group:** The sample size is set to $M_1$, our theoretically derived lower bound for achieving the target performance with high probability.

- **Conservative Group:** The sample size is set to $M_2$, a more conservative bound, which should comfortably meet or exceed the target performance.

For each group, we trained a RandomForestRegressor model and evaluated its performance using Mean Absolute Error (MAE), Root Mean Squared Error (RMSE), and the coefficient of determination ($R^2$). We also evaluated the data coverage using our proposed integrated coverage metric, which combines k-NN, adaptive grid, and manifold-based approaches. The experiment was repeated over 10 trials with different random seeds to ensure the statistical robustness of our findings.

The key parameters for the experiment were set as follows:

- **Target MAE ($\varepsilon_{target}$):** We set a target performance of 0.5 for the Mean Absolute Error.

- **Confidence Level ($\delta$):** The confidence parameter was set to 0.05, implying a 95% confidence that the true MAE is within our target.

- **Intrinsic Dimension ($d^*$):** Estimated from the data, yielding $d^* \approx 6.03$.

- **Lipschitz Constant ($L_{est}$):** Estimated using a Ridge model, resulting in $L_{est} \approx 0.45$.

- **Derived Epsilon ($\varepsilon$):** Calculated as $\varepsilon = \varepsilon_{target}/L_{est} \approx 1.10$.

Based on these parameters, the theoretical sample complexity bounds were calculated as:

- $M_1$ (Sufficient) = 7,494

- $M_2$ (Conservative) = 106,212

The corresponding sample sizes for the experimental groups were 749, 7,494, and 106,212.

The experimental results strongly support our theoretical framework. The average performance metrics across 10 trials for each group are summarized in Table 1 and visualized in Figure 7.

Table 1: Mean Performance Metrics Across Experimental Groups

| Group | Sample Size | MAE | RMSE | R² |
|-------|-------------|-----|------|-----|
| Insufficient | 749 | 0.440 | 0.632 | 0.692 |
| Sufficient | 7,494 | 0.379 | 0.566 | 0.755 |
| Conservative | 106,212 | 0.361 | 0.546 | 0.772 |

As predicted, we observe a clear trend of performance improvement with increasing sample size. The MAE for the **Sufficient** group (0.379) is significantly better than the **Insufficient** group (0.440) and successfully meets our target MAE of 0.5. The **Conservative** group shows a marginal improvement over the Sufficient group, suggesting that the $M_1$ bound is already effective at capturing the required sample size.

Figure 8 further details the relationship between sample size and performance. The plots show a clear trend of diminishing returns, where performance gains (lower MAE, higher $R^2$) and coverage rate saturate as the sample size increases towards the 'Conservative' level. This saturation effect underscores the efficiency of the $M_1$ bound, as it achieves most of the potential performance gains with a fraction of the data required by the $M_2$ bound.

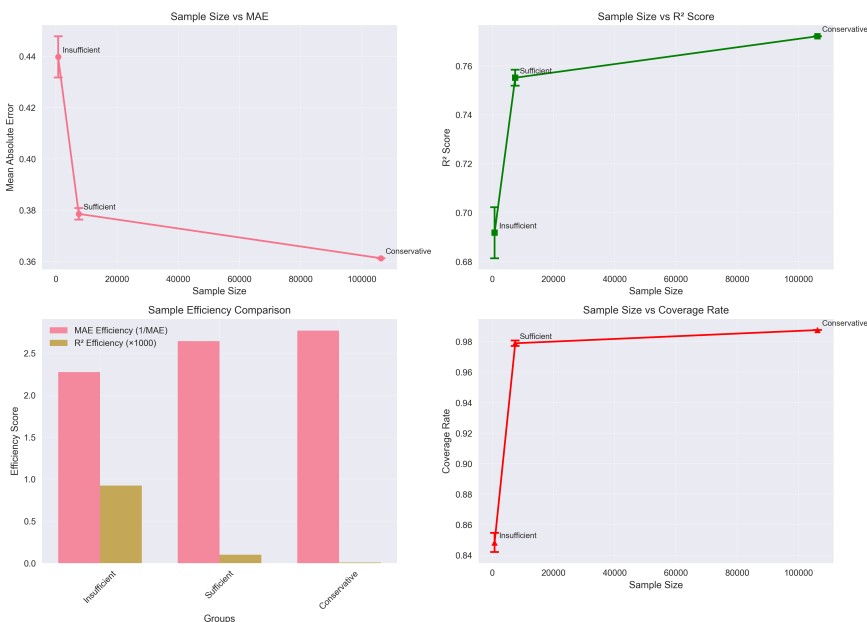

Figure 7: Comprehensive performance comparison across the three experimental groups. The bar chart (top-left) and box plot (top-right) illustrate the improvement in MAE and $R^2$ with larger sample sizes. The scatter plots show the trade-off between sample size and MAE (bottom-left) and coverage versus $R^2$ (bottom-right).

Figure 8: Analysis of sample efficiency. The plots show a clear trend of diminishing returns, where performance gains (lower MAE, higher $R^2$) and coverage rate saturate as the sample size increases. The efficiency comparison (bottom-left) highlights how the 'Sufficient' group offers a good balance between performance and data cost.

Statistical analysis confirms these observations. An ANOVA test revealed a statistically significant difference in MAE, RMSE, and R² across the three groups ($p < 0.001$ for all metrics). Subsequent Tukey's HSD post-hoc tests showed that all pairwise comparisons between the groups were significant. Figure 9 visualizes these results, where a value of "1" (red) indicates a statistically significant difference ($p < 0.05$) between the groups. This confirms that each increase in sample size tier led to a meaningful performance gain.

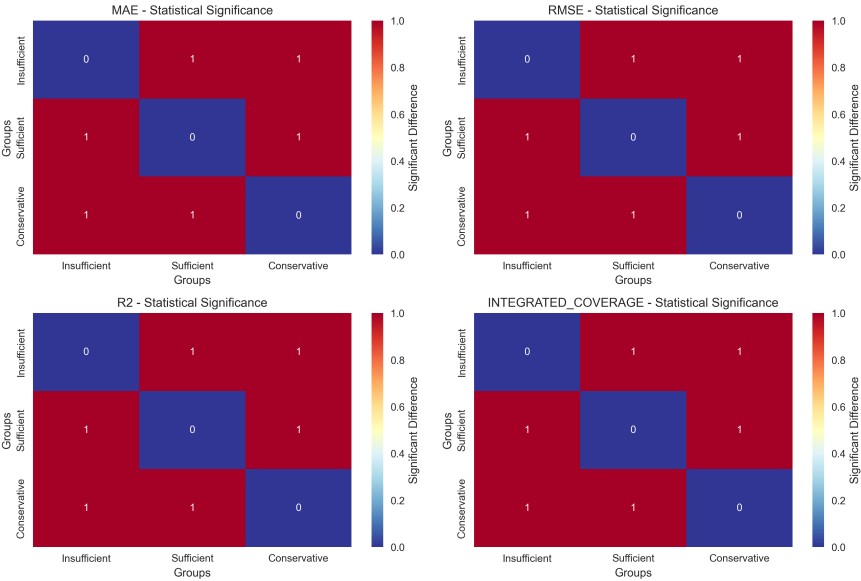

Figure 9: Heatmaps of statistical significance from Tukey's HSD post-hoc tests. The value "1" (red) indicates a statistically significant difference ($p < 0.05$) between the groups, while "0" (blue) indicates no significant difference. All pairwise comparisons for MAE, RMSE, R², and Integrated Coverage are shown to be significant.

In conclusion, the experimental results on the California Housing dataset provide strong empirical evidence for the validity of our theoretical sample complexity bounds. The experiment demonstrates that our framework can effectively estimate the required sample size to achieve a predefined performance target in a practical machine learning application, with visualizations and statistical tests confirming the significance of our proposed sample size tiers.

## 5 Broader Impact

Our result supports principled, resource-aware sampling strategies for verification tasks that require uniform guarantees over continuous domains. Positive impacts include reduced computational burden and a clearer calibration of confidence vs. cost. Potential risks arise if assumptions (uniformity, independence, Lipschitz structure) are violated in deployment; we recommend diagnostics to detect distribution shift or non-uniform sampling. Environmental benefits follow from fewer samples at high confidence, reducing energy costs.

## 6 Conclusion and Future Work

This study establishes a tight lower bound of $M = O(\tilde{C} \ln(\frac{2\tilde{C}}{\delta}))$ for complete coverage in uniformly sampled $d$-dimensional unit cubes, representing a fundamental improvement over classical coupon collector bounds. Our analysis leverages spatial discretization and Chebyshev-based probabilistic techniques to achieve logarithmic dependence on failure probability, compared to the classical linear dependence. This improvement is particularly significant for high-confidence scenarios and high-dimensional problems, where classical bounds become overly conservative. The theoretical results are validated through comprehensive numerical experiments, demonstrating practical relevance for algorithm verification and resource allocation. Future work

will explore tighter bounds using advanced concentration inequalities and extend the framework to adaptive sampling schemes and broader applications in machine learning and optimization.

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

## A  Appendix: Proof of Theorem 1

To derive the lower bound of sample complexity, we first apply the unilateral Chebyshev inequality:

$$P(Z - E[Z] \geq a) \leq \frac{Var[Z]}{a^2}.$$

Let $a = kE[Z]$, satisfying $(k + 1)E[Z] = 1$. Assuming $E[Z] < 1$, when $1 - E[Z] > 0$, substituting into the inequality yields:

$$P(Z \geq 1) \leq \frac{Var[Z]}{(kE[Z])^2} = \frac{Var[Z]}{(1 - E[Z])^2}. \tag{5}$$

The requirement $P(Z \geq 1) \leq \delta/2$ is given by (5), that is:

$$\frac{Var[Z]}{(1 - E[Z])^2} \leq \frac{\delta}{2}. \tag{6}$$

From Lemma 1 and (3):

$$E[Z] = \tilde{C}(1 - \frac{1}{\tilde{C}})^M. \tag{7}$$

For any subcube $i$, its uncovered indicator variable $Y_i$ satisfies:

$$E[Y_i] = (1 - \frac{1}{\tilde{C}})^M, \quad Var[Y_i] = E[Y_i](1 - E[Y_i]).$$

Since the expression for $Z$ is:

$$Z = \sum_{i=1}^{\tilde{C}} Y_i.$$

Therefore, the variance of $Z$ can be obtained as:

$$Var[Z] = \sum_{i=1}^{\tilde{C}} Var[Y_i] + \sum_{i \neq j} Cov(Y_i, Y_j).$$

Meanwhile, calculate $Cov(Y_i, Y_j)$:

$$\begin{aligned}
Cov(Y_i, Y_j) &= E[Y_i Y_j] - E[Y_i] \cdot E[Y_j] \\
&= (1 - \frac{2}{\tilde{C}})^M - (1 - \frac{1}{\tilde{C}})^{2M} \\
&= \frac{1}{\tilde{C}^{2M}} \cdot (\tilde{C}^M (\tilde{C} - 2)^M - (\tilde{C} - 1)^{2M}) < 0.
\end{aligned} \tag{8}$$

Then it can be obtained that $Cov(Y_i, Y_j) < 0$, so the upper bound of variance is:

$$Var[Z] = \sum_{i=1}^{\tilde{C}} Var[Y_i] + \sum_{i \neq j} Cov(Y_i, Y_j) \leq \sum_{i=1}^{\tilde{C}} Var[Y_i] = \tilde{C} Var[Y_i]. \tag{9}$$

Substitute equations (7) and (9) into (6):

$$\frac{\tilde{C} E[Y_i](1 - E[Y_i])}{(1 - \tilde{C} E[Y_i])^2} \leq \frac{\delta}{2}. \tag{10}$$

Let $q = E[Y_i] = (1 - \frac{1}{\tilde{C}})^M$, and (10) transforms to:

$$\frac{\tilde{C} q (1 - q)}{(1 - \tilde{C} q)^2} \leq \frac{\delta}{2}.$$

Expand and organize to obtain the quadratic inequality:

$$q^2 (2\tilde{C} + \delta \tilde{C}^2) - 2\tilde{C} q (1 + \delta) + \delta \geq 0. \tag{11}$$

Solve the inequality (11), whose discriminant is:

$$\Delta = 4\tilde{C}^2 + 8\tilde{C}\delta(\tilde{C} - 1). \tag{12}$$

Since $\tilde{C} \geq 1$ and $\delta > 0$, $\Delta > 0$. The quadratic equation has two real roots:

$$q_1 = \frac{2\tilde{C}(1 + \delta) - \sqrt{\Delta}}{2(2\tilde{C} + \delta \tilde{C}^2)}, \quad q_2 = \frac{2\tilde{C}(1 + \delta) + \sqrt{\Delta}}{2(2\tilde{C} + \delta \tilde{C}^2)}. \tag{13}$$

Since $a = 2\tilde{C} + \delta \tilde{C}^2 > 0$, the inequality solves to $q \leq q_1$ or $q \geq q_2$. Analysis shows that $q_2 \geq 1$, so we take $q \leq q_1$. Substituting $q = (1 - \frac{1}{\tilde{C}})^M$ for $q \leq q_1$ and taking the logarithm gives:

$$M \geq \tilde{C} \ln \frac{1}{q_1} \geq \frac{\ln q_1}{\ln(1 - \frac{1}{\tilde{C}})}. \tag{14}$$

Therefore, we can choose $M = O(\tilde{C} \ln \frac{1}{q_1})$. To establish this bound, we first derive the asymptotic approximation for $q_1$ when $\delta \to 0^+$. Firstly, we analysis the expansion of $\Delta$ (12):

$$\Delta = 4\tilde{C}^2 + 8\tilde{C}\delta(\tilde{C} - 1). \implies \sqrt{\Delta} = 2\tilde{C}\sqrt{1 + 2\delta(1 - \frac{1}{\tilde{C}})}.$$

By a Taylor expansion

$$\sqrt{1 + x} = 1 + \frac{x}{2} - \frac{x^2}{8} + O(x^3),$$

we can obtain that

$$\sqrt{\Delta} = 2\tilde{C}(1 + \delta(1 - \frac{1}{\tilde{C}}) - \frac{\delta^2}{2}(1 - \frac{1}{\tilde{C}})^2 + O(\delta^3)). \tag{15}$$

Substitute equation (15) into (13)

$$
\begin{aligned}
q_1 &= \frac{2\tilde{C}(1 + \delta) - \sqrt{\Delta}}{2(2\tilde{C} + \delta\tilde{C}^2)} \\
&= \frac{2\tilde{C}(1 + \delta) - 2\tilde{C}(1 + \delta(1 - \frac{1}{\tilde{C}}) - \frac{\delta^2}{2}(1 - \frac{1}{\tilde{C}})^2 + O(\delta^3))}{2(2\tilde{C} + \delta\tilde{C}^2)} \\
&= \frac{2\delta + \tilde{C}\delta^2(1 - \frac{2}{\tilde{C}}) + O(\delta^3)}{4\tilde{C}(1 + \frac{\delta\tilde{C}}{2})}.
\end{aligned}
$$

Then we calculate the ratio:

$$
\begin{aligned}
\frac{q_1}{\frac{\delta}{2\tilde{C}}} = q_1 \cdot \frac{2\tilde{C}}{\delta} &= \frac{4\tilde{C} + 2\tilde{C}^2\delta(1 - \frac{2}{\tilde{C}}) + O(\delta^2)}{4\tilde{C}(1 + \frac{\delta\tilde{C}}{2})} \\
&= \frac{1 + \frac{\delta\tilde{C}}{2}(1 - \frac{2}{\tilde{C}}) + O(\delta^2)}{1 + \frac{\delta\tilde{C}}{2}} \\
&= \frac{1 + O(\delta)}{1 + O(\delta)}.
\end{aligned}
$$

Such that, when $\delta \to 0^+$

$$\lim_{\delta \to 0^+} \frac{q_1}{\frac{\delta}{2\tilde{C}}} = \lim_{\delta \to 0^+} \frac{1 + O(\delta)}{1 + O(\delta)} = 1.$$

Finally, we prove that

$$q_1 \approx \frac{\delta}{2\tilde{C}}.$$

Substitute (14) to obtain the final lower bound:

$$M = O(\tilde{C}\ln(\frac{2\tilde{C}}{\delta})).$$

## B    Experiment Appendix

### B.1    Hardware Configuration

The experiments were conducted on a workstation with the following specifications:

- **CPU:** AMD Ryzen® 9 9950X3D @ 4.3 GHz base / 5.7 GHz boost

- **Memory:** 96 GB DDR5-5600

- **GPU:** NVIDIA® GeForce RTX™ 5090 32 GB GDDR7

### B.2    Experimental Environment for Parameter Analysis of Grid Coverage

### B.2.1    Software Environment

The software stack used for the experiments is detailed below:

- **Operating System:** Windows 10 Pro (64-bit)

- **Python Version:** 3.12.3

- **Core Libraries:**
  - numpy: 1.26.4
  - matplotlib: 3.9.0
  - seaborn: 0.13.2
  - psutil: 5.9.8

### B.2.2 Dataset: Synthetic Grid Coverage Simulation

- **Source:** This experiment does not use an external dataset. It is a pure simulation based on the `GridCoverage` class, which models the problem of covering a d-dimensional grid with randomly sampled points.

- **Scale:** The simulation runs up to a maximum of 1,000,000 samples (`max_samples`) for each parameter configuration to determine the point at which full grid coverage is achieved.

- **Preprocessing:** Not applicable, as the experiment is a direct simulation of a mathematical model.

### B.2.3 Parameter Settings

The experiment systematically analyzes the impact of three key parameters by varying one while keeping the others fixed. The ranges and fixed values are:

- **Dimension (d):**
  - Varied values: [1, 2, 3]
  - Fixed value (when not varied): 2

- **Precision (epsilon):**
  - Varied values: np.linspace(0.05, 0.5, 10)
  - Fixed value (when not varied): 0.1

- **Failure Probability (delta):**
  - Varied values: np.linspace(0.02, 0.2, 10)
  - Fixed value (when not varied): 0.1

- **Lipschitz Constant (L_tilde):** Fixed at 1.0 for all simulations.

- **Number of Trials (num_trials):** 10 trials are run for each parameter setting to ensure statistical robustness.

### B.2.4 Running Conditions

- The script leverages multiprocessing to run experimental trials in parallel, significantly reducing the total execution time. It is designed to utilize all available CPU cores.

- The simulation can be memory-intensive, especially for higher dimensions, but is generally manageable with 32 GB of RAM.

- The primary output consists of three enhanced analysis plots (`dimension_enhanced_analysis.png`, `epsilon_enhanced_analysis.png`, `delta_enhanced_analysis.png`), which are saved to disk. Ensure write permissions are available.

### B.3 Experimental Environment for DCDC Sample Complexity Validation

#### B.3.1 Software Environment

The software stack used for the experiments is detailed below:

- **Operating System:** Windows 10 Pro (64-bit)

- **Python Version:** 3.12.3

- **Core Libraries:**
  - torch: 2.3.1
  - numpy: 1.26.4
  - matplotlib: 3.9.0
  - seaborn: 0.13.2
  - tqdm: 4.66.4

#### B.3.2 Dataset: Synthetic Ornstein-Uhlenbeck Process

- **Source:** The data is synthetically generated on-the-fly by simulating a 2-dimensional Ornstein-Uhlenbeck (OU) process. This avoids reliance on external datasets and allows for precise control over the data-generating dynamics.

- **Scale:** For each experimental group (Insufficient, Sufficient, Conservative), a set of points is sampled uniformly from the domain. The number of samples is determined by the theoretical sample complexity calculations ($M_{theory}$ and $M_{classic}$).

- **Preprocessing:** No explicit preprocessing is required as the data is generated in a controlled manner. The training process involves sampling mini-batches from a fixed set of uniformly distributed points within the specified bounds.

#### B.3.3 Parameter Settings

The key parameters for the DCDC experiment are defined in the `ExperimentConfig` class:

- **System Parameters:**
  - `dimension`: 2
  - `epsilon`: 0.05 (error tolerance)
  - `delta`: 0.01 (failure probability)
  - `lipschitz_constant`: 1.0

- **Ornstein-Uhlenbeck Process:**
  - `beta`: 4.0 (mean reversion rate)
  - `sigma`: 4.0 (volatility)
  - `dt`: 0.01 (time step)

- **Training Parameters:**
  - `max_epochs`: 50,000
  - `learning_rate`: 7e-5
  - `batch_size`: 256

- **Network Architecture (`MLPNetwork`):**
  - `hidden_dims`: [512, 512, 256] with Tanh activation.

### B.3.4 Running Conditions

- A CUDA-enabled GPU is highly recommended for this experiment due to the computational demands of training the deep neural network over many epochs.

- The script is self-contained and generates all necessary data.

- The experiment saves comprehensive results, including JSON files with detailed metrics, loss curves, 3D surface plots of the learned value function, and a summary report in Markdown format. Ensure write permissions are available in the execution directory.

## B.4 Experimental Environment for California Housing Sample Complexity Validation

### B.4.1 Software Environment

The software stack used for the experiments is detailed below:

- **Python Version:** 3.12.3
- **Core Libraries:**
  - numpy: 1.26.4
  - pandas: 2.2.2
  - scikit-learn: 1.5.0
  - umap-learn: 0.5.6
  - scipy: 1.13.1
  - matplotlib: 3.9.0
  - seaborn: 0.13.2
  - statsmodels: 0.14.2
  - plotly: 5.22.0

### B.4.2 Dataset: California Housing

- **Source:** The experiment utilizes the California Housing dataset, available through the `scikit-learn` library. This dataset is based on data from the 1990 California census.

- **Scale:** The dataset consists of 20,640 samples and 8 numerical features. The target variable is the median house value for California districts.

- **Preprocessing:** The features are first standardized using `StandardScaler` to have zero mean and unit variance. Subsequently, a whitening transformation is applied to the scaled data to remove correlations between features and ensure they have a common variance. The data is then split into training (80%) and testing (20%) sets.

### B.4.3 Parameter Settings

The key parameters for the California Housing experiment are defined in the `ExperimentConfig` class:

- **Target MAE (`target_mae`):** 0.5. This defines the desired precision for the regression model, which is used to derive the error tolerance $\epsilon$.

- **Failure Probability (`delta`):** 0.05. This is the acceptable probability of the sample complexity bounds not holding.

- **Random State (`random_state`):** 42. Used for ensuring reproducibility in data splitting and model training.

- **Number of Trials (`n_trials`):** 10. The number of Monte Carlo simulations to run for robust statistical analysis.

- **Test Set Size (`test_size`):** 0.2. Specifies that 20% of the data is reserved for testing.

### B.4.4 Running Conditions

- The experiment is designed to be self-contained and does not require external data downloads beyond what is provided by `scikit-learn`.

- The execution involves multiple Monte Carlo trials, which can be computationally intensive and time-consuming.

- The script generates and saves several output files, including detailed JSON results, statistical analysis reports, and performance visualizations (`.png`, `.html`). Ensure write permissions are available in the execution directory.

