# OpenReview forum: "Improved Sample Complexity for Full Coverage in Compact and Continuous Spaces via Probabilistic Analysis"
_TMLR — Rejected by TMLR_

### Review · Reviewer_NARk · 2025-10-18

**Summary Of Contributions:**

The paper studies a problem arising naturally in a large class of machine learning settings, where one is required to discretize a given compact domain into $d$-dimensional unit hypercubes and obtain uniformly random samples from these in order to provide high-confidence bounds. This problem naturally falls within the framework of the coupon collector problem, or its related variations. However, as the authors point out, it seems that the complexity bounds obtained in the literature in the context of ``continuous-domain, Lipschitz continuous functions'' typically rely on Markov’s inequality, which scales with $\frac{1}{\delta}$, where $\delta$ is the failure probability.

The clear issue with using Markov’s inequality is that it does not capture the fact that the number of uncovered cubes has a sharp concentration around its mean, and moreover, that the covered events are also combinatorially structured. In contrast, the argument presented takes advantage of the exponential form of the occupancy probability. This, along with the fact that Lipschitz continuity of the objective function ensures that the variations inside each subcube are controlled, leads to a bound that scales with $\log(\frac{1}{\delta})$.

**Audience:**

Yes

**Audience Explanation:**

The authors list a large number of problems in the literature for which the obtained results would be important, for example Deep Contraction Drift Calculators.

**Broader Impact Concerns:**

Broader Impacts are addressed and there are no concerns.

**Claims And Evidence:**

Yes

**Claims Explanation:**

As mentioned, the argument presented takes advantage of the exponential form of the occupancy probability. This, along with the fact that Lipschitz continuity of the objective function ensures that the variations inside each subcube are controlled, leads to a bound that scales with $\log(\frac{1}{\delta})$ with $\delta$ the failure probability. The proof of the main claim in the paper follows relatively standard arguments in the literature on occupancy problems and appears to be correct to me. Even though it is a structural re-use of a relatively well-known probabilistic threshold in a new setting, it provides the right asymptotics for the supremum error in continuous domains, which appears to be missing from the literature.

**Requested Changes:**

Now, I believe that a simple union bound already provides sufficiency, an upper bound on the probability, that scales with $\log(\frac{1}{\delta})$. To provide a tight bound, however, one needs a more careful argument. As done in the classical literature on occupancy problems (as indicated in the paper, for instance, in~Feller1991), the authors use a Chebyshev-type argument, which accounts for second moments. The proof is relatively straightforward and appears to be correct to me. In light of what I just mentioned about the tightness of the bound, should we not obtain a $\Theta(C \log(C/\delta))$ result?

Given that the main contribution of the paper is to point out a missing observation in the literature, I also wonder whether the extensive simulation results, while perhaps interesting, are necessary and have added value.

---

### Review · Reviewer_H65Y · 2025-10-20

**Summary Of Contributions:**

Motivated by uniformly verifying inequalities over continuous spaces, this paper presents a better bound for the probability of "not all sub cubes are covered". I think it is a theoretically meaningful and technically correct contribution to the stats/ML literature. The presentation can be (and should be) improved, I think. Please see below.

**Audience:**

No

**Audience Explanation:**

Using neural networks to approximate some functions that satisfy some inequalities (e.g., Lyapunov functions) is part of AI for science, where uniformly verifying inequalities over continuous spaces is essential to validate the approximation, so (part of) TMLR's audience should be interested in this paper.

Update (yes to no): Please see above.

**Claims And Evidence:**

No

**Claims Explanation:**

I go through the proof line by line, and it is correct. (There seems to be a typo in the expression of q_1 on page 15, but it does affect correctness.)

Update (yes to no): I thank reviewer u3yd for pointing out that the "classical coverage analyses" part of the claim is wrong. Given the existence of an elementary proof of the logarithmic dependence, the contribution of this paper becomes quite limited. Although there seems to be some detour in the proof, I appreciate the authors' efforts in it.

**Requested Changes:**

I would like to recommend for acceptance after a revision improving the presentation.

The problem studied in this paper (uniformly verifying inequalities over continuous spaces) is a general one, and DCDC is an example of application. Therefore, I suggest introducing the problem in general (Section 1-3) and discussing how it can be applied to DCDC and other settings in separate sections. Currently, the problem introduction is in terms of DCDC, which has quite some extra structures that are not essential to the problem (and may confuse readers). For example, $KV-V+U$ can be replaced by a single function (e.g., $V$), avoiding undefined $KV$ and $\hat{K}_NV$. This will also make it clearer that only the set $M$ is random. By assuming this single $V$ is $L$-Lipschitz, the $|KV(x)-KV(y)|$ part in Section 3 (where $f$ is undefined and unnecessary to formulate the general problem) can be removed. In summary, I think it is much cleaner to have a general formulation in Section 3 (a Lipschitz function + subcubes + uniform samples) and bring in DCDC later (which requires quite some space to set up).

Other minor suggestions:
1. Lemma 1 can be merged into the proof of Theorem 1.
2. For the three assumptions on page 4, I think they are just settings, not assumptions.
3. The "via probabilistic analysis" part in the title may be removed, as it is a general term.
4. Some parenthesis (e.g., the ones in Theorem 1, $M=O(\cdot)$) are too small.

---

### Review · Reviewer_u3yd · 2025-11-23

**Summary Of Contributions:**

This paper studies the number of i.i.d. uniform samples in $[0,1]^{d}$ required to cover all $\tilde{C}$ equal-sized subcubes with probability at least $1-\delta$. The paper claims that "classical" approach yields a sample complexity that scales linearly with $1/\delta$, and that their approach improves this to $\log 1/\delta$.

Unfortunately, this core claim is **wrong**. In fact, even a direct application of Markov’s inequality already yields a logarithmic dependence on $1/\delta$, not a linear one.

To see this, let $Z$ denote the number of uncovered subcubes after drawing $M$ i.i.d. uniform samples, using the paper's notation. Simple calculation shows that $\mathbb{E}(Z) =\tilde{C} (1 - \frac{1}{\tilde{C}} )^M$. Applying Markov’s inequality to the nonnegative random variable $Z$ yields $\mathbb{P}(Z \geq 1)\leq \mathbb{E}(Z) $. Moreover, using $1-x\leq e^{-x}$,
$$
\mathbb{E}(Z)  = \tilde{C} (1 - \frac{1}{\tilde{C}} )^M \leq \tilde{C} e^{-M/\tilde{C}}.
$$
Therefore, in order for $\mathbb{P}(Z\geq 1)\leq \delta$, it is sufficient to let $\tilde{C} e^{-M/\tilde{C}} \leq \delta$, which leads to $M\geq \tilde{C} \log (\frac{\tilde{C} }{\delta})   $.

**Audience:**

No

**Audience Explanation:**

Given that the paper’s main claim is incorrect as explained above, I do not believe the findings would be of interest to TMLR’s audience.

**Broader Impact Concerns:**

No concerns.

**Claims And Evidence:**

No

**Claims Explanation:**

The main claim in the paper is wrong as explained above.

**Requested Changes:**

Because the paper's main claim is wrong, no incremental adjustments would move it toward acceptance.

---

> ### Author Response · Authors · 2025-11-24
> **Response to Reviewer u3yd**
>
> Dear Reviewer u3yd,
>
> Thank you for your rigorous review. Your assessment that "the core claim is wrong" has helped us recognize a critical ambiguity in how we framed our work.
>
> ---
>
> ## Clarifying our main claim
>
> **Our main claim was not that classical coupon collector methods inherently yield only linear $1/\delta$ dependence.** Nor did we claim novelty for the $\log(1/\delta)$ rate itself, which is well established. Rather, **our claim is that DCDC's Theorem 5 employs stopping-time analysis yielding $M = O(\tilde{C}\log \tilde{C}/\delta)$ (linear in $1/\delta$), whereas our count-based analysis yields $M = O(\tilde{C}\log(\tilde{C}/\delta))$ (logarithmic).**
>
> Your derivation showing that applying Markov’s inequality to $Z$ achieves $O(\log(1/\delta))$ is correct and fully aligned with our method. The key difference is that DCDC adopts a stopping-time framework rather than this count-based approach.
>
> We acknowledge that "classical linear $1/\delta$ dependence" without specifying DCDC's stopping-time analysis created confusion, and we apologize for this lack of clarity.
>
> ---
>
> ## DCDC's stopping-time analysis
>
> DCDC's proof states [1]:
>
> > "It is well-known that we need $\Theta(\tilde{C} \log \tilde{C})$ on average to collect $\tilde{C}$ different coupons. By Markov inequality, we can choose $M = O(\tilde{C} \log \tilde{C}/\delta)$ to reduce the failure probability below $\delta/2$."
>
> This analyzes stopping time $T$ via $\Pr(T > M) \leq \mathbb{E}[T]/M$, yielding linear $1/\delta$ dependence.
>
> Your derivation analyzes uncovered count $Z$: $\mathbb{E}[Z] \le \tilde{C} e^{-M/\tilde{C}}$, $\Pr(Z \ge 1) \le \mathbb{E}[Z]$, yielding $M \ge \tilde{C} \log(\tilde{C}/\delta)$—logarithmic dependence. Our work adopts this perspective within DCDC's framework, quantifying up to 98.5% sample reduction.
>
> ---
>
> ## Variance analysis value
>
> Building on count-based analysis, we prove $\mathrm{Cov}(Y_i,Y_j) < 0$ and $\mathrm{Var}[Z] < \mathbb{E}[Z]$. Via Chebyshev with explicit constant
> $$
> q_1 = \frac{2\tilde{C}(1+\delta) - \sqrt{4\tilde{C}^2 + 8\delta \tilde{C}(\tilde{C}-1)}}{2(2\tilde{C} + \delta \tilde{C}^2)},
> $$
> we obtain significantly tighter bounds in practice. Across three scenarios, the ratio between the actual required samples and our variance-based bound remains $0.72$–$0.80$, whereas for the DCDC stopping-time bound this ratio is below $0.1$.
>
> ---
>
> ## Planned revisions
>
> 1. Remove "classical linear $1/\delta$" phrasing; specify baseline as DCDC's stopping-time analysis.
> 2. State upfront we contrast with DCDC (not broader literature) and claim no novelty for $\log(1/\delta)$ itself.
> 3. Expand Related Work distinguishing stopping-time vs. count-based; cite your derivation.
> 4. Reframe contributions: count-based analysis in DCDC framework (98.5% reduction), variance tightening (ratio $\approx 0.75$ vs. $<0.1$), cross-domain validation.
>
> ---
>
> We hope that by clarifying our main claim and implementing these revisions, the relationship between our work, DCDC’s analysis, and the classical coupon collector literature will be much clearer.
>
> ---
> [1] Qu et al., NeurIPS (2024): Deep Learning for Computing Convergence Rates of Markov Chains.

---

> > ### Comment · Reviewer_u3yd · 2025-11-26
> > **Response to authors**
> >
> > Thank you for the clarification that your comparison was intended only for a specific theorem in a specific paper. However, as written, the title, abstract, and several parts of the main text repeatedly frame the contribution as an improvement to the sample complexity bounds for covering subcubes with uniform samples. Correcting this would require a substantial revision of the paper’s scope and framing. Even with such corrections, the contribution would remain minimal: the $\log(1/\delta)$ dependence is well-known and the derivation is elementary; and the fact that one earlier work stated a looser bound does not justify novelty for this paper.

---

### Decision · Action_Editor_Qefj · 2026-01-14

**Recommendation:** Reject

**Audience:**

Yes

**Audience Explanation:**

The better convergence rates for the DCDC method would be interesting for researchers in this subarea but the scope is rather limited.

**Claims And Evidence:**

No

**Claims Explanation:**

As pointed out by Reviewer u3yd, the current submission over-claims its contribution in a general problem setting about the sample complexity bounds for covering subcubes with uniform samples where the dependency on $\log(1/\delta)$ is well established. The actual contribution is to provide a better bound for a specific theorem in a specific paper (DCDC). It requires a substantial revision to the manuscript to clarify the claims.